# TASK MATRICES: LINEAR MAPS FOR CROSS-MODEL FINETUNING TRANSFER

## ABSTRACT

Results in interpretability suggest that large vision and language models learn implicit linear encodings when models are biased by in-context prompting. However, the existence of similar linear representations in more general adaptation regimes has not yet been demonstrated. In this work, we develop the concept of a task matrix, a linear transformation from a base to finetuned embedding state. We demonstrate that for vision and text models and ten different datasets, a base model augmented with a task matrix achieves results surpassing linear probes, sometimes approaching finetuned levels. We show that linear encoding in transformer embedding spaces exists between pretrained and finetuned architectures, and can be readily exploited through task matrices. These matrices incur low computational costs, and are both data-efficient and generalizable in multiple domains. We make our implementation publicly available.

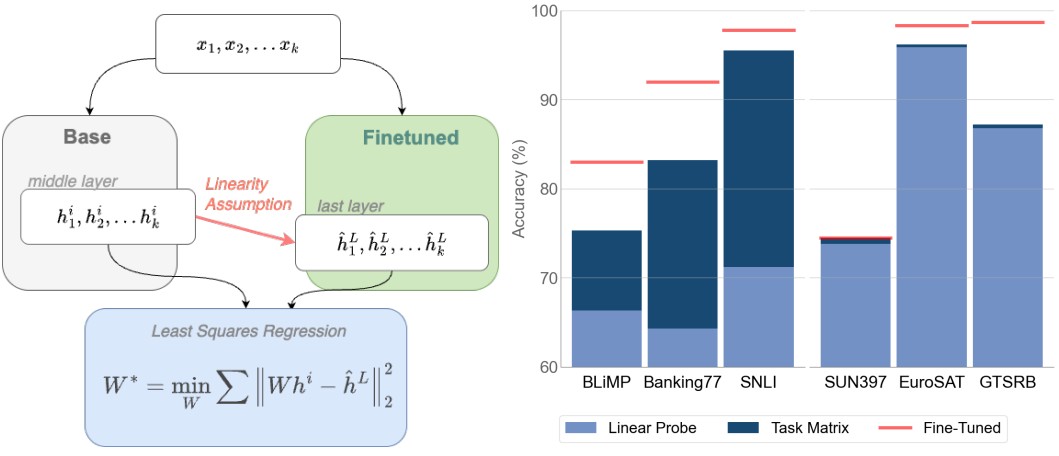

Figure 1: **Left:** On many datasets, employing a linearity assumption between base and finetuned model states offers lightweight and effective approximations. **Right:** Applying a task matrix beats linear probes, and sometimes reaches finetuned performance.

## 1 INTRODUCTION

Domain adaptation is a fundamental challenge for practitioners using foundation models, who often wish to leverage a pretrained model for a specific downstream task. In order to accomplish domain adaptation robustly and effectively, finetuning model layers have been the traditional approach for improving downstream performance on specialized datasets (Devlin et al., 2018).

While the pretrain-and-finetune paradigm underlies crucial adaptation techniques such as reinforcement learning with human feedback, adapting a pretrained model poses substantial barriers in both training time and compute resources for practitioners. In recent years, there has been increasing interest in developing low-memory and parameter-efficient alternatives to finetuning. Prominent

methods include linear probes and low-rank adaptation (LoRA) (Alain & Bengio, 2018; Hu et al., 2021).

In this work, we employ a concept learning hypothesis to develop a novel method for transferring fine-tuned performance to base models. First introduced by Paccanaro & Hinton (2001), linear transformations between vector representations have been found to be effective for relational approximation between given concepts. In the transformer architecture setting, Hernandez et al. (2023) demonstrated that model architectures often employ near-linear transformations over relations in the setting of **in-context learning**. However, in many cases, model adaptation is accomplished instead through finetuning. This leads us to the following line of inquiry:

*Does linear representation work effectively over adaptation by model finetuning?*

Like in-context learning, finetuning adapts a base model to a specific setting, drawing upon existing representations within a specialized domain. However, rather than inducing biases solely by modifying context, finetuning alters model-internal representations. Accordingly, finetuning can lead either to preservation of transferred concepts (Yosinski et al., 2014; Zhang & Wu, 2024) or representational collapse (Aghajanyan et al., 2020). Based on interpretability results highlighting representational flexibility in middle layers, we introduce the concept of the **task matrix**:

---

A **task matrix** is an $N_{\text{embed}} \times N_{\text{embed}}$ linear transformation from a base model representation to a fine-tuned representation, where the finetuned model has been trained on a dataset $D$. This task matrix is built upon a **linearity assumption**. Specifically, we propose that a linear map $W$ transforms the hidden representation at a fixed intermediate layer of a base model, $x \in H_{\text{base}}$, into the last-layer representation of the finetuned model $y \in H_{\text{ft}}$:

$$Wx \approx y$$

The task matrix is then constructed through regression over samples from $D$, on pairs of base and finetuned hidden representations.

---

Multiplying base embeddings by a task matrix then produces an approximation of the finetuned output, which is passed to downstream head(s) for decoding.

We make the following core observations:

- **Task matrices improve model performance on diverse tasks, outperforming probing baselines**. RoBERTa augmented with task matrices beat linear probes by as much as 80% on challenging multi-class datasets, and often comes within a percentage of finetuned performance.

- **Linear relationships between base and fine-tuned models are readily exploitable.** Task matrices can be constructed with as little as 1% of the training data, and are an effective way to compress information learned by finetuned models. In vision, finetuned models exhibit *increasing decodability* in later layers, while in text, performance often peaks in middle layers.

- **Task matrices generalize over multiple tasks, and are robust to reductions in data.** Extending the linearity assumption to joint datasets, we find that vision task matrices can be employed for multiple tasks with marginal decreases in individual accuracies, dropping from 92% to 81% from 1 to 8 classification datasets. Moreover, task matrices exhibit relative improvements in data-scarce settings against both probe baselines and finetuned models.

Overall, our work demonstrates that in both vision and text settings, linear relationships exist between pretrained and finetuned model layers. We demonstrate that these relationships are learnable from small quantities of representative data, and generalize to multiple datasets. This results in a novel adaptation technique, which can be used when storing or releasing finetuned models is impractical or commercially infeasible.

## 2 RELATED WORK

### 2.1 LINEAR APPROXIMATION OF CONCEPT MAPPINGS

Within concept learning, relationships between vector encodings have long been represented as matrix transformations, for instance in representing hierarchical data structures and models of compositional semantics (Paccanaro & Hinton, 2001; Coecke et al., 2010).

A substantial body of interpretability literature has subsequently provided evidence for linear representations of concepts within model architectures (Mikolov et al., 2013; Elhage et al., 2022; Park et al., 2024). Linear representation has likewise been utilized to identify concepts and modify predictions through hidden representation interventions (Hernandez et al., 2023; Chanin et al., 2024; Xia & Kalita, 2025). We take inspiration from the setup and hypotheses of these works, especially **middle state enrichment** found by Geva et al. (2021).

However, unlike the prior works, which consider either in-context learning or intra-model linearity, we demonstrate linear representations between pretrained and finetuned models. This allows us to posit a general paradigm of linear mapping over adaptation. This framework is applicable to many downstream applications, without being subject to particular relational constraints.

### 2.2 LINEAR OPERATIONS IN TRANSFORMERS

Based on the observation that enriched subject states contain significant attribution information in early-intermediate layers, intervention-based approaches to factual editing have emerged. This family of techniques take advantage of concept orthogonality in pretrained models (Meng et al., 2023b;a; Fang et al., 2025). In our work, we pursue a similar goal of isolating concept maps; however, we pursue a different objective, that of domain adaptation from base representations.

### 2.3 EXPLOITING LINEAR STRUCTURE FOR IMPROVED PERFORMANCE

Parameter-efficient tuning methods, such as LoRA, have emerged around low-rank hypotheses over the adaption process (Hu et al., 2021; Zhang et al., 2023; Zaken et al., 2021; Li & Liang, 2021). Targeted intervention based on linear hypotheses has likewise been performed in a single-model setting by Yom Din et al. (2023), improving on direct layer readouts as seen in LogitLens (nostalgebraist, 2020). Here, we extend prior work to make linear assumptions between two different models, employing a targeted intervention for layer-specific representations.

## 3 APPROACH

### 3.1 PRELIMINARIES & LINEARITY ASSUMPTION

We focus on transformer architectures, which have seen state-of-the-art results across vision, text, and multimodal tasks (Vaswani et al., 2017). We briefly mention relevant details from the standard architecture below. Let the initial embedding be $h^0 \subset H^0 \in \mathbb{R}^d$, where $d$ is the hidden dimension. The embedding is then updated by $L$ transformer blocks, such that for each $\ell \in [1, L]$,

$$h^\ell \subset H^\ell = b^\ell(h^{\ell-1}), \text{ where } b^\ell = b^{\text{ffn},\ell} \circ b^{\text{attn},\ell}$$

is a composition of multi-head self-attention and feed-forward layers, typically including residual connections and layer normalization. The final representation $h^L$ is then projected from $H^L \in \mathbb{R}^d$ to a task-specific space in $\mathbb{R}^N$, by a finetuned classification head $V$.

Our linearity assumption is as follows. For convenience, let the finetuned model's last-layer space be $H_{\text{ft}}$, and the base model's output space at a fixed layer $i \in \{1, 2, \ldots, L\}$ be $H_{\text{base}}$. Let a sample population from the dataset be $\{x_1 \ldots x_k\} \in D$, so that $X \in \mathbb{R}^{k \times d}$ and $Y \in \mathbb{R}^{k \times d}$ are $k \times d$ matrices of base and finetuned representations. We assume that for some layer $i$, there exists a matrix $W \in \mathbb{R}^{d \times d}$ such that for all pairs $(x, y) \in H_{\text{base}} \times H_{\text{ft}}$:

$$Wx \approx y$$

This assumption is motivated by Geva et al. (2023)'s analysis of factual recall in LMs, where **enriched subject representations** containing key attributes are found *prior to the last layer representation*.

We suggest that during the process of model adaption, the finetuned model learns to approximate output classifications across a task-specific dataset $D$ primarily through interpreting these existing representations. In particular, the richer set of intermediate-layer attributes suggest that decoding from these layers may outperform adaptation strategies from the final layer.

## 3.2 TASK MATRIX CONSTRUCTION

Let $S = \{x_1, \ldots x_k\}$ be a dataset over which a base (pretrained) model has been finetuned. Let $h_k^i \in \mathbb{R}^d$ and $\hat{h}_k^L \in \mathbb{R}^d$ denote, respectively, the $i^{\text{th}}$ layer base representation and the final-layer representation of its finetuned counterpart. Let the number of output classes be $N$, and the final trained decoder head be $V \in \mathbb{R}^{d \times N}$. We posit there exists a linear transformation $W \in \mathbb{R}^{d \times d}$ such that for $k \in \{1, \ldots n\}$, $Wh^i$ will result in a faithful approximation of $\hat{h}^L$:

$$\arg \max_N V(Wh^i) = \arg \max_N V(\hat{h}^L)$$

We would like to estimate $W$ with an approximation $W^*$. To do this, we minimize the least-squares loss between the embeddings over $x_1, \ldots x_n$: [1] [2]

$$\mathcal{L}_{\text{LS}}(W) = \frac{1}{S} \sum_{k=1}^{S} \left\| Wh_k^i - \hat{h}_k^L \right\|_2^2$$
$$W^* = \arg \min_{W \in \mathbb{R}^{d \times d}} \mathcal{L}_{\text{LS}}(W).$$

At inference time, for a test sample $j$, we compute

$$\tilde{h}_j^L = W^* h_j^i,$$

and use $\tilde{h}_j^L$ in place of the final finetuned representation.

## 3.3 LINEAR PROBE BASELINE

We employ a familiar technique as a comparable baseline, linear probing. Given a specialized dataset, a linear probe (or adapter) re-trains the decoder head, and is thus comparable in runtime to linear regression over a $d \times d$ matrix. For weight matrix $V$ and vector $b$, the linear operation applied to the final-layer embedding $h_k^L$ is:

$$z_k = V h_k^L + b \in \mathbb{R}^N, \qquad V \in \mathbb{R}^{N \times d}, \ b \in \mathbb{R}^N.$$

Both the task matrix and linear probe optimize over training samples $x_1, \ldots, x_k$. In contrast to the linear approximation, in which we minimize $|h_k^i - \hat{h}_k^L|$, the linear probe has the more traditional objective of minimizing misclassification of the final output, $z_k$. Values for $V$ and $b$ are obtained by minimizing the standard cross-entropy loss $\mathcal{L}_{CE}(z)$ over the dataset $z_1 \ldots z_k$:

$$(V^*, b^*) = \arg \min_{V \in \mathbb{R}^{d \times N}, \ b \in \mathbb{R}^N} \mathcal{L}_{\text{CE}}(z).$$

Task matrix comparison to linear probing can be seen in Tables 1 and 2.

## 4 METHODOLOGY

Our experiments focused on architectures with sufficient depth, as shallow networks demonstrated reduced approximation efficacy. To select datasets for task matrix construction, we prioritized well-known and popular datasets for which existing benchmarks exist. We then selected datasets exhibiting substantial performance gaps between base and fine-tuned models. This filter allowed for meaningful evaluation of the finetuned approximation over a baseline of the pretrained model.

---

[1]We also experimented with Ridge regression (Hoerl & Kennard, 1970) which can result in improved solutions for multi-collinear datasets but did not find improvements over the baseline method. Other approximation techniques, including those seen in Hernandez et al. (2023), are promising avenues for future work.

[2]$W^*$ can be accurately learned with very few samples, in some cases <10. In 5.3, we demonstrate that this robust regression property allows for superior relative performance in limited data settings.

In order to create a task matrix, it is necessary to define the representations on which the linearity holds. For both text and vision architectures, we utilized `[CLS]` tokens as unified representations over which task matrices are calculated. We extract `[CLS]` tokens from intermediate layers of the base model and the final layer of the fine-tuned model: for samples $x_1, \ldots x_j$, this is $h_j^i$ and $\tilde{h}_j^L$ respectively.

For the sample population $x_1, \ldots x_j$, in practice all training images for a dataset $D_{\text{train}} = \{x_1, x_2, \ldots x_n\}$ were used; see the Appendix for experiments with a subset of training images. After computing the task matrix mapping a base layer to the final fine-tuned layers for $D_{\text{train}}$, the task matrix transformation was applied back on the base layer for $D_{\text{test}}$, and evaluated using the fine-tuned classifier head.

### 4.1 VISION

For image classification, we selected a multi-layer transformer architecture which has produced state-of-the-art results, the CLIP ViT B-32 (Radford et al. (2021)) Vision Tower. We did not use the text encoder, and trained an end-to-end classification network on the vision component alone. Full CLIP ViT B-32 results with only the vision encoder fine-tuned and models used from Tang et al. (2024) are in Appendix C.3.

We constructed task matrices for the following datasets: DTD, EuroSAT, GTSRB, MNIST, RESISC45, Stanford Cars, SUN397, and SVHN (Cimpoi et al., 2014; Helber et al., 2019; Stallkamp et al., 2012; LeCun et al., 2010; Cheng et al., 2017; Krause et al., 2013; Xiao et al., 2010; Netzer et al., 2011). The datasets encompass diverse classification tasks, including texture recognition, scene categorization, vehicle identification, digit classification, and traffic sign detection. See Appendix G for vision dataset sourcing.

To model conditions in deployment, both finetuning and linear probes were performed until no further improvement. This was typically many more epochs than the task matrix, which learns from $\leq 1$ iteration of the training data. Over the eight datasets, this method consistently achieved close to state-of-the-art results.

### 4.2 TEXT

To simplify experiments, we focused on sentence representations, which immediately led to readily approximable patterns between base and finetuned models. We adapted the masked language model RoBERTa for sentence classification through examining `[CLS]` token representations. We also used a standard sentence transformer architecture, all-Mini-LM-v2; results can be found in Appendix B.

We evaluated the models across seven diverse NLP benchmarks. These include Emotion (Saravia et al., 2018) for emotion classification in Twitter messages, HANS (McCoy et al., 2019) for testing natural language inference heuristics and BLiMP (Warstadt et al., 2020). To extend BLiMP for classification, we treated minimal pairs from the 67 grammatical phenomena categories as sentence-level classification problems. We also tested TREC-6 (Li & Roth, 2002) for question classification; SNLI (Bowman et al., 2015) for natural language inference; ATIS (Hemphill et al., 1990) for intent detection in flight information queries; and Banking-77 (Casanueva et al., 2020) for fine-grained banking intent classification. See Appendix F for detailed dataset descriptions.

## 5 RESULTS

Below, we show the efficacy of task matrices at exploiting non-final layer linearities, demonstrate they are robust to data-constrained and multi-task settings, and validate their causal influence to predictions. The majority of results are averaged over five independent runs (n=5) and reported with a 95% confidence interval (95% CI).

### 5.1 TASK MATRIX PERFORMANCE

We find the strongest results for RoBERTa, outperforming linear probes from the same data distribution on all seven tested datasets. On the vision side, we find similar results and often come within a percentage point of finetuned accuracy, while linear probes also perform well.

| Method | Emotion | HANS | BLiMP | Trec-6 | SNLI | ATIS | Banking77 |
|---|---|---|---|---|---|---|---|
| (classes) | (6) | (2) | (67) | (6) | (3) | (18) | (77) |
| Linear Probe | 58.9±1.5 | 81.3±0.8 | 66.3±1.5 | 79.8±1.5 | 71.2±0.5 | 89.3±0.2 | 64.3±3.0 |
| Task Matrix | **66.0±2.8** | **96.8±0.2** | **75.3±1.0** | **84.9±1.2** | **76.3±1.8** | **95.5±0.3** | **83.2±1.4** |
| (best layer) | (1,2,10) | (16) | (4,5,6) | (11) | (17) | (4,6) | (1,3,4) |
| Fine-Tuned | 91.4±0.8 | 100.0±0.0 | 83.0±1.0 | 95.1±1.6 | 88.7±0.6 | 97.8±0.3 | 92.0±0.7 |

Table 1: Task Matrix against text baselines (%), RoBERTa-large (n=5, 95% CI). Layers are zero-indexed.

| Method | DTD | EuroSAT | GTSRB | MNIST | RESISC | Stanford Cars | SUN397 | SVHN |
|---|---|---|---|---|---|---|---|---|
| (classes) | (47) | (10) | (43) | (10) | (45) | (196) | (397) | (10) |
| Linear Probe | **77.2±0.3** | 95.9±0.1 | 86.8±0.1 | 98.7±0.1 | **91.7±0.2** | **79.9±0.2** | 73.8±0.3 | 66.6±0.3 |
| Task Matrix | 75.7±0.5 | **96.2±0.4** | **87.2±0.3** | **99.03±0.1** | 89.1±0.6 | 79.7±0.5 | **74.8±0.3** | **66.7±0.7** |
| (best layer) | (11) | (6,8) | (11) | (7,8) | (11) | (11) | (11) | (8) |
| Fine-Tuned | 77.4±1.4 | 98.3±0.5 | 98.7±0.1 | 99.4±0.1 | 92.3±0.5 | 82.7±0.5 | 74.5±0.3 | 96.4±0.2 |

Table 2: Task Matrix against vision baselines (%), CLIP ViT-B/32 vision tower (n=5, 95% CI). Layers are zero-indexed.

We also show results for all-MiniLM-L12-v2, DeIT, and DINOv3 in the Appendix sections B, D. and E, respectively, demonstrating our approach is generalizable across models (Wang et al., 2020; Touvron et al., 2020; Siméoni et al., 2025).

### 5.2 LAYER-BY-LAYER PERFORMANCE

To better understand matrix performance, we produce layer-wise graphs (Figure 2) of the best-performing approximation for each base model layer. Each layer represents a **different linearity hypothesis**, so that task matrix performance reflects the **linear decodability** of the base model space $K_{\text{base}}$ in $K_{\text{ft}}$.

In vision, we observe an upward trend for task matrix performance across all layers, demonstrating increasing decodability over all layers. We note that CLIP-SUN397 and CLIP-Stanford Cars performance demonstrate low decodability throughout early and middle layers, rapidly rising only at late layers. These datasets exhibit the highest class counts (397 and 196 respectively), suggesting fine-grained classification boundaries can reduce intermediate linearities.

In contrast, text task matrices often perform best at intermediate layers, and performance remains relatively stable throughout [3]. For instance, performance peaks at layer 18 for RoBERTa-SNLI and layer 7 for RoBERTa-BLiMP. This suggests enriched representations develop prior to the final layer readout which are readily mappable. Differing from the vision side once more, there exists no clear relationship between the strength of intermediate linearities and category counts.

### 5.3 TASK MATRICES IN DATA-SCARCE SETTINGS

We next investigate the robustness of the task matrix's linear approximation in settings where the majority of data is held out for both probes and the task matrix. Concretely, we finetune the model on a 20% split of the training data, and subsequently construct a task matrix with the same quantity of restricted data and a linear probe with the same quantity of restricted data. We find that task matrices are far more robust to changes in data quantity than linear probes, exhibiting a 82% improvement on ATIS and 81% improvement on Trec-6 (Table 3). In Appendix C.1, we show similar results for CLIP.

---

[3]The notable exception is SNLI, a natural language inference task for predicting entailment. Performance steadily increases in early and middle layers, and decreases in later layers.

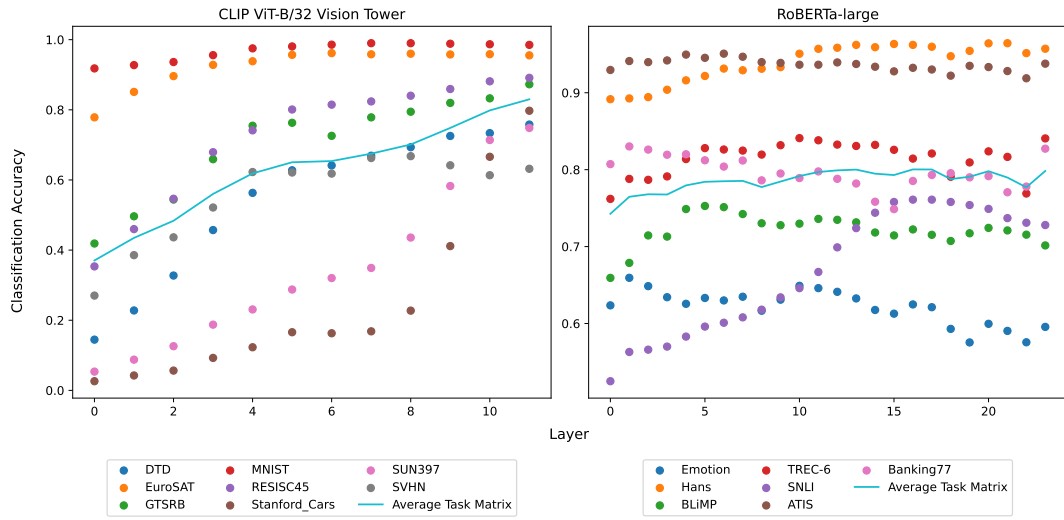

Figure 2: Best layer-wise performance of CLIP and RoBERTa task matrices on respective datasets. Average of five trials.

| Method | Emotion | HANS | BLiMP | Trec-6 | SNLI | ATIS | Banking77 |
|---|---|---|---|---|---|---|---|
| (classes) | (6) | (2) | (67) | (6) | (3) | (18) | (77) |
| Linear Probe | 25.8±2.8 | 81.5±2.3 | 26.6±10.1 | 30.3±6.8 | 59.1 | 42.7±2.4 | 14.1±1.5 |
| Task Matrix | **36.3±2.1** | **95.8±0.4** | **55.2±1.6** | **55.0±3.2** | **76.4** | **77.7±1.9** | **46.5±1.1** |
| (best layer) | (1,2) | (16) | (4,5,6) | (11) | (18,19) | (4,5,6) | (3,4) |
| Fine-Tuned | 63.9±1.6 | 100.0±0.0 | 70.2±1.9 | 76.7±5.0 | 87.1 | 91.9±1.6 | 79.0±1.1 |

Table 3: Task Matrix against text baselines (%) with training samples limited to 20% of the original dataset. The results exhibit minimal relative differences from the full training results in Table 1. RoBERTa (n=5, SNLI n=2, 95% CI). Layers are numbered 0-23.

## 5.4 TASK MATRICES FOR MULTITASK CLASSIFICATION

We further investigate whether a *single* task matrix can exist for *multiple* datasets, as done by Ilharco et al. (2022) for model weight arithmetic. To formulate the task matrix for the multi-dataset domain, we replace our original linearity hypothesis with a joint assumption on linearity. Extending our original notion of concept representation, we instead posit that a transformation in model space can benefit multiple datasets. The task matrix then learns a joint mapping to an optimal space for all datasets. This means that the final layer embeddings $\hat{h}^L$ are sampled from a joint dataset $N = \{S_1, S_2, \ldots S_n\}$, while the base embeddings remain unchanged:

$$(h^i, \hat{h}^L) = (h^i, \bigcup_{S \in N} \hat{h}^L)$$

We then create task matrices following the technique outlined in Section 3.2 with the total number of samples equal to the sum of all samples used to train the fine-tuned models of each selected dataset. To evaluate task matrices across the selected datasets $\{d_1, ..., d_n\}$ on the test sample $j$, we multiply the same task matrix $W^*$ with the intermediate representation $h^j$, and pass results through the respective fine-tuned classification heads $D_1, \ldots D_n$ to obtain predictions.

As seen in Figure 3, which represents multi-task task matrices performance on all pairs of datasets $D_i \times D_j$, matrices maintain performance on both targeted tasks, validating the hypothesis above. Quantitative normalized accuracy is shown in Section C.2 in the Appendix.

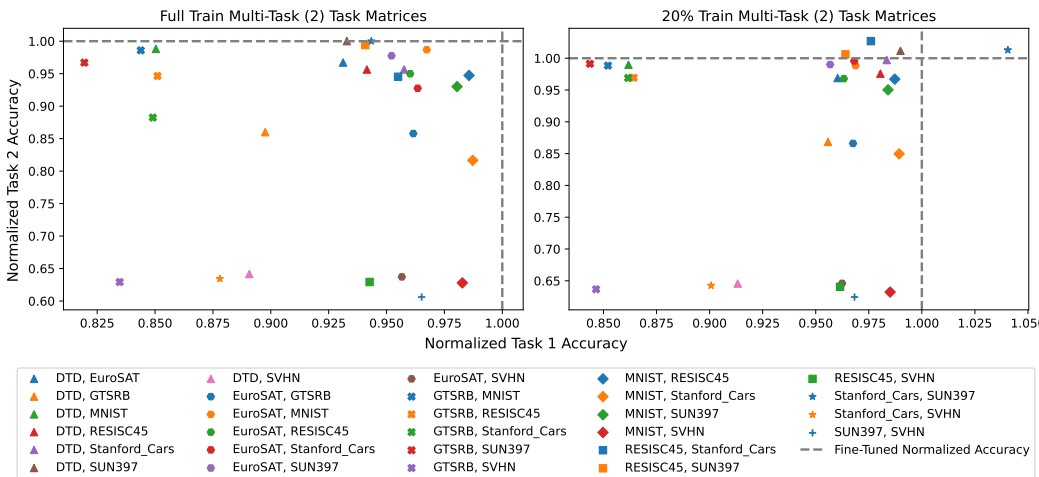

Figure 3: CLIP ViT-B/32 Vision 2 Task Augmentation. Learned linear approximations are beneficial for each dataset, and exhibit relative improvements in the data-scarce setting.

We further show multi-task performance in Figure 4. A single task matrix across multiple datasets remains highly effective over individual evaluations.

## 5.5 Ablations: Frozen Base Classifier Head & Finetuned Decoding

Below, we perform two ablation studies to further show the efficacy of the task matrix.

**Ablation: Direct Readout from Base Model**

One potential confounding factor with the methodology we developed is determining whether task matrix performance arises from transformation or simply from the fine-tuned classifier head. To isolate these effects, we conducted a controlled ablation experiment, in which we test the base model representations with a fine-tuned classifier head alone. As seen in Tables 4 and 5, the **Base w/ FT Classifier** method performs worse than task matrices on all datasets across all settings. By effectively replacing task matrices with the identity, the ablation demonstrates the necessity of the transformation for improved performance.

**Ablation: Frozen Decoder Head**

To validate that our approach does not rely on adapted components, we modify our technique to operate on a frozen decoder head, which means our technique does not require any finetuned model components. In Table 6, we show that vision results with a frozen decoder head closely match earlier performances in Table 2 This establishes that employing a task matrix alone is **sufficient** for good performance.

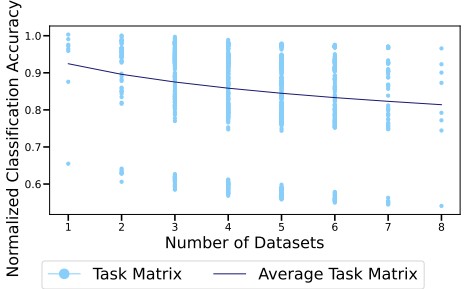

Figure 4: CLIP ViT Vision Full Train Multi-Task (1-8) Layer 11 Results. Small drops in accuracy are seen (from 92%-81%). (n=5).

| Base w/ FT Classifier Method | Emotion | HANS | BLiMP | Trec-6 | SNLI | ATIS | Banking77 |
|---|---|---|---|---|---|---|---|
| (classes) | (6) | (2) | (67) | (6) | (3) | (18) | (77) |
| Full Train | 25.5±13.6 | 50.1±0.2 | 2.4±0.8 | 23.1±0.7 | 33.7±0.7 | 18.6±13.8 | 1.7±0.2 |
| (best layer) | (23) | (23) | (23) | (23) | (23) | (23) | (23) |

Table 4: RoBERTa Base w/FT Classifier Ablation performance (%). The classifier head was taken from the fine-tuned model and used to directly read out from the base model. Task matrices as seen in Tables 1 and 3 greatly exceed the results here, demonstrating the task matrix is necessary for the improved performance. RoBERTa (n=5, 95% CI). Layers are numbered 0-23.

| Base w/ FT Classifier Method | DTD | EuroSAT | GTSRB | MNIST | RESISC | Stanford Cars | SUN397 | SVHN |
|---|---|---|---|---|---|---|---|---|
| (classes) | (47) | (10) | (43) | (10) | (45) | (196) | (397) | (10) |
| Full Train | 59.4±1.4 | 65.5±5.4 | 45.5±5.7 | 48.9±1.5 | 73.6±2.4 | 53.8±1.8 | 65.3±1.1 | 24.3±2.2 |
| (best layer) | (11) | (10,11) | (11) | (11) | (11) | (11) | (11) | (11) |
| 20% Train | 50.8±2.3 | 61.7±2.9 | 47.1±6.1 | 39.3±24.8 | 70.4±2.7 | 36.5±2 | 56.3±1 | 23.3±3 |
| (best layer) | (11) | (11) | (11) | (11) | (11) | (11) | (11) | (11) |
| Frozen Head | 3.1±0.9 | 14.8±4.8 | 4.09±1.8 | 34.1±19.8 | 14.3±4.3 | 0.7±0.07 | 0.3±0.04 | 17.8±2.2 |
| (best layer) | (8,9,11) | (0,2,5,8) | (1,7,10,11) | (5,7,10,11) | (0,2,4,6,9) | (4,7,8,10) | (6,10) | (0,6,7,8,10) |

Table 5: Vision Base w/FT Classifier Ablation performance (%). The classifier head was taken from the fine-tuned model and used to directly read out from the base model. Task Matrix performance in Tables 2, and 8 exceed results here in all datasets, demonstrating that the task matrix is significant. CLIP ViT-B/32 vision tower (n=5, 95% CI). Layers are numbered 0-11.

| Method | DTD | EuroSAT | GTSRB | MNIST | RESISC45 | Cars | SUN397 | SVHN |
|---|---|---|---|---|---|---|---|---|
| (num. classes) | (47) | (10) | (43) | (10) | (45) | (196) | (397) | (10) |
| Task Matrix | 74.9±0.4 | 96.6±0.3 | 86.9±0.4 | 99±0.08 | 90.3±0.5 | 72±0.5 | 70.1±0.4 | 67.7±0.8 |
| (best layer) | (11) | (6,7,9) | (11) | (7) | (11) | (11) | (11) | (8) |
| Fine-Tuned | 75.9±0.7 | 98.4±0.6 | 99.01±0.1 | 99.4±0.06 | 94±0.7 | 78.6±1.2 | 66.5±0.1 | 96.3±0.1 |

Table 6: Frozen Decoder Head performance (%). The classifier head was randomly initialized and frozen while fine-tuning. CLIP ViT-B/32 vision tower (n=5). Layers are numbered 0-11.

## 6 CONCLUSION

Recent results in interpretability suggest that models contain linear substructure, in particular under input-output constraints such as object prediction from relational examples. In this work, we apply linear representation hypotheses to the broader problem of domain adaptation, positing that the representational changes which result from gradient-based fine-tuning likewise employ linear readouts from early layers.

With this theoretical justification, we introduce task matrices as linear mappings between base and fine-tuned model states that improve the performance of a base model on specialized datasets. We find that while performance varies in effectiveness across datasets, such a transformation can result in competitive performance with the specialized model itself. Experiments show that these transformations exist across a range of tasks, including sentiment classification, image recognition, and natural language inference. We observe further that these transformations can learn a range of tasks while retaining high individual accuracy, and that they are robust to reduced data regimes.

ACKNOWLEDGMENTS

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

## A    TRAINING DATA AND MODEL EFFICACY

Across both vision and text regression, we observed double descent, a phenomena which has been studied in depth in prior literature (Bach, 2023; Bartlett et al., 2020; Mei & Montanari, 2022; Schaeffer et al., 2024). As seen in Figure 5, the task matrix performance rises with the number of images used in construction, but declines sharply near the full dimension of the embedding space (768 in CLIP). When the number of input samples is equal to the embedding dimension, a unique exact solution exists. In practice, we employed many more images than the embedding dimension, opting to use the full training dataset for task matrix construction.

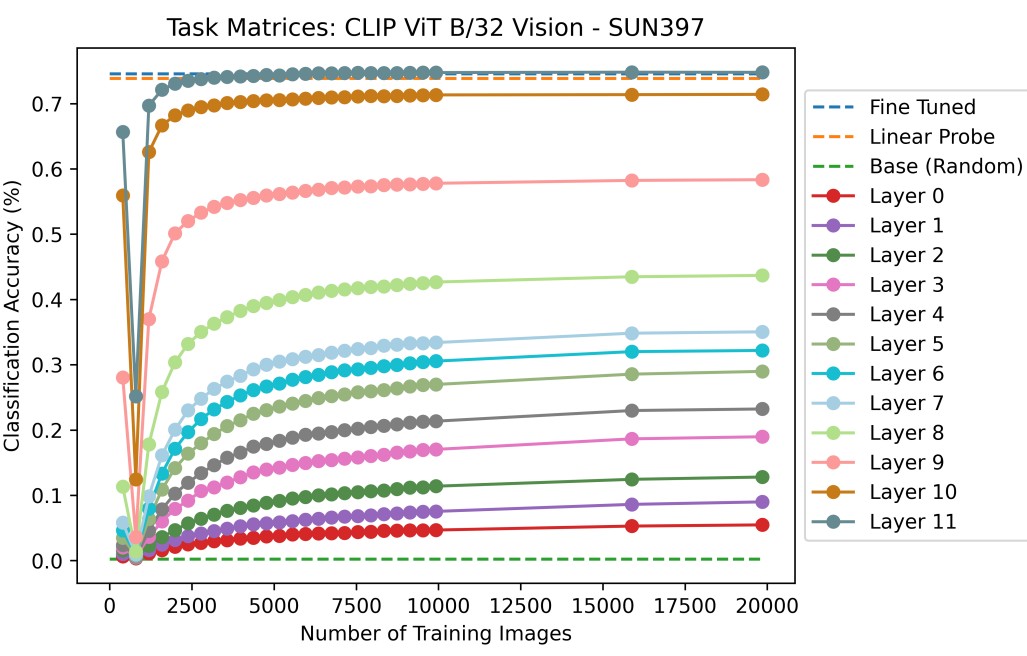

Figure 5: Classification Accuracy of SUN397 Across Layers and Training Images.

## B    ALL-MINILM-L12-V2: SENTENCE TRANSFORMER RESULTS

| Method | Emotion | HANS | BLiMP | Trec-6 | SNLI | ATIS | Banking 77 |
|---|---|---|---|---|---|---|---|
| (classes) | (6) | (2) | (67) | (6) | (3) | (18) | (77) |
| Base w/ FT Classifier | 24.5±13.0 | 50.4±0.7 | 2.6±0.8 | 22.5±1.5 | 33.6±0.3 | 21.6±36.3 | 23.2±4.4 |
| Linear Probe | **66.8±0.6** | 76.0±0.8 | 38.1±1.7 | 75.1±1.3 | 56.7±0.3 | **94.9±0.6** | **90.9±0.9** |
| Task Matrix | 63.5±1.0 | **82.3±2.0** | **50.0±5.0** | **84.7±0.9** | **64.5±0.2** | 91.9±0.7 | 88.3±1.2 |
| (best layer) | (L11) | (L8) | (L3,4) | (L0,5) | (L7) | (L5) | (L9,10) |
| Fine-Tuned | 81.7±1.4 | 99.4±1.2 | 60.5±3.6 | 93.2±0.5 | 85.1±0.1 | 93.5±0.4 | 89.0±1.0 |

Table 7: Task Matrix against text baselines (%), all-MiniLM-L12-v2 (n=5, 95% CI). Layers are zero-indexed

# C CLIP ViT-B/32: Additional Results

## C.1 Data-Scarce Results

| Method | DTD | EuroSAT | GTSRB | MNIST | RESISC | Stanford Cars | SUN397 | SVHN |
|---|---|---|---|---|---|---|---|---|
| (classes) | (47) | (10) | (43) | (10) | (45) | (196) | (397) | (10) |
| Linear Probe | **67.2±1** | 94.6±0.3 | 84.3±0.4 | 98.2±0.1 | 88.2±0.3 | **62.8±0.5** | 66.8±0.3 | 62.5±0.6 |
| Task Matrix (best layer) | 61.9±0.7 (11) | **95.5±0.7** (6,7,9) | **85.8±0.3** (11) | **98.9±0.1** (7,8) | **89.7±0.3** (11) | 50.9±1 (11) | **68.1±0.2** (11) | **64.5±0.8** (8) |
| Fine-Tuned | 67.5±1 | 97.5±0.4 | 97.8±0.2 | 99.3±0.1 | 91.5±0.5 | 58.2±1 | 67±0.3 | 93.6±0.3 |

Table 8: Task matrix performance against vision baselines (%) with training samples limited to 20% of the original dataset. The results here exhibit minimal differences from the full training results in Table 2. CLIP-ViT-B/32 vision tower (n=5, 95% CI). Layers are zero-indexed.

## C.2 Quantitative 2-Task Accuracy

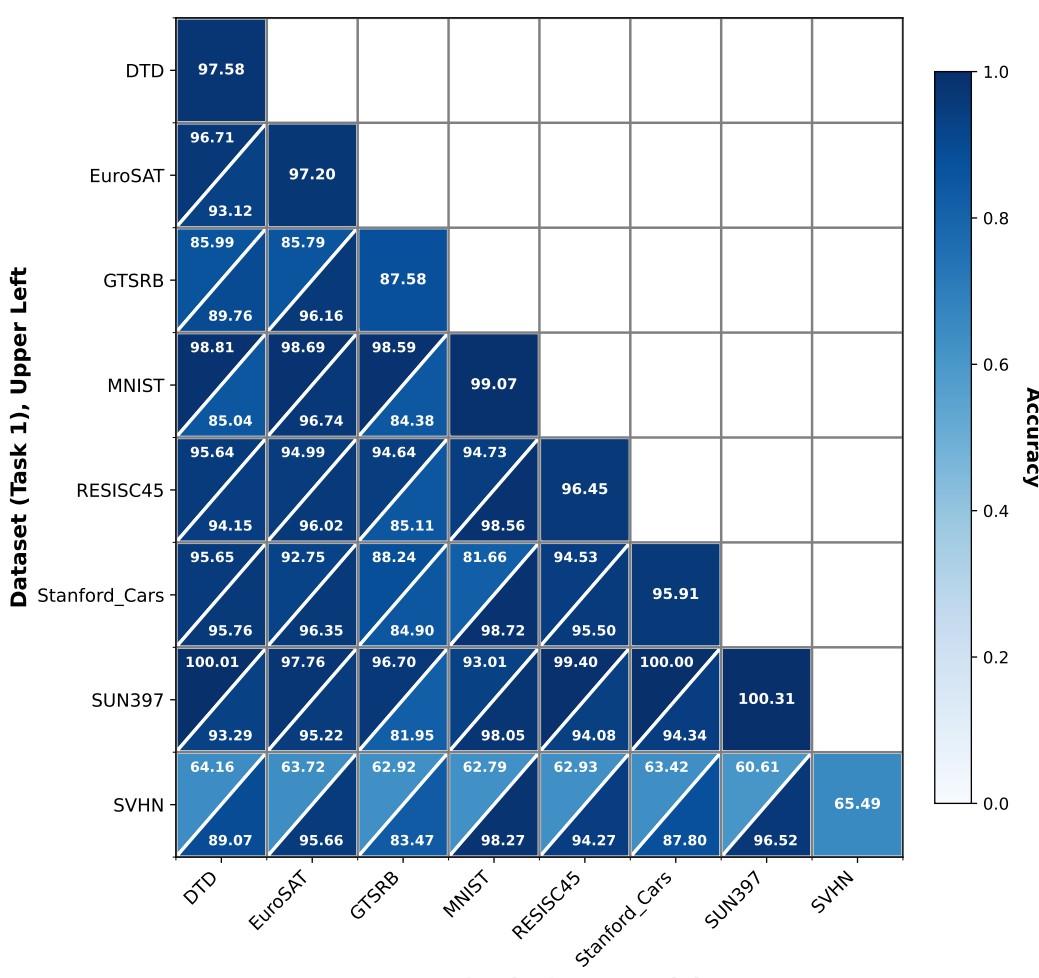

Figure 6: Average Normalized 2-Task Classification Accuracy of CLIP ViT B/32 (n=5). Last Layer.

## C.3 Full CLIP ViT B-32 Results

We next test CLIP ViT B-32 on open vocabulary classification with the frozen text encoder. We replaced the vision model of the base CLIP with models available from Tang et al. (2024) for all 8 vision classification datasets. Results demonstrate that Task Matrices reach competitive performance in frozen-text encoder settings.

| Method | DTD | EuroSAT | GTSRB | MNIST | RESISC | Stanford Cars | SUN397 | SVHN |
|---|---|---|---|---|---|---|---|---|
| (classes) | (47) | (10) | (43) | (10) | (45) | (196) | (397) | (10) |
| Base | 40.79 | 42.11 | 27.57 | 30.75 | 51.01 | 57.95 | 60.4 | 14.76 |
| Task Matrix | **72.48** | **96.51** | **84.82** | **97.34** | **88.03** | **71.5** | **69.98** | **60.84** |
| (best layer) | (11) | (7) | (11) | (9) | (11) | (11) | (11) | (8) |
| Fine-Tuned | 76.38 | 98.66 | 97.69 | 99.32 | 94.26 | 76.79 | 71.52 | 95.93 |

Table 9: Task matrix performance against vision baselines (%). The results here exhibit fairly minimal differences from the full training results in Table 2. CLIP-ViT-B/32 (n=1). Layers are zero-indexed.

## D DeiT: Comprehensive Vision Results

| Method | DTD | EuroSAT | GTSRB | MNIST | RESISC | Stanford Cars | SUN397 | SVHN |
|---|---|---|---|---|---|---|---|---|
| (classes) | (47) | (10) | (43) | (10) | (45) | (196) | (397) | (10) |
| Base w/ FT Classifier | 54.8±1.6 | 70.4±3 | 30±5.5 | 47.1±9.8 | 57.7±1.7 | 5.6±1 | 41.8±0.8 | 23.8±3 |
| Linear Probe | **63.3±0.3** | 93.4±0.05 | **66±0.2** | 95.6±0.08 | **79.1±0.2** | 26.3±0.1 | **49.3±0.3** | 46.5±0.5 |
| Task Matrix | 62.5±0.6 | **95.1±0.2** | 64.9±1.1 | **95.7±0.4** | 77.9±0.3 | **31.4±0.4** | 48.9±0.1 | **49.6±1** |
| (best layer) | (L10,11) | (L5) | (L7) | (L4,5) | (L9) | (L9) | (L11) | (L7) |
| Fine-Tuned | 67.4±0.6 | 98±0.2 | 96.7±0.5 | 99.2±0.1 | 89.3±0.2 | 50.7±1.1 | 55.1±0.3 | 95.1±0.2 |

Table 10: Task matrix against vision baselines (%), DeiT-tiny-patch16-224 (n=5, 95% CI). Layers are zero-indexed.

| Method | DTD | EuroSAT | GTSRB | MNIST | RESISC | Stanford Cars | SUN397 | SVHN |
|---|---|---|---|---|---|---|---|---|
| (classes) | (47) | (10) | (43) | (10) | (45) | (196) | (397) | (10) |
| Base w/ FT Classifier | 42.6±1.2 | 71.4±1.7 | 34.6±3.9 | 43±10.4 | 55.9±1.8 | 4.4±0.6 | 31.7±0.5 | 22.7±3.5 |
| Linear Probe | **52.1±1.5** | 91.6±0.1 | 62.8±0.4 | 95.1±0.1 | 73.1±0.3 | **12.4±0.2** | **38.7±0.5** | 45.1±0.4 |
| Task Matrix | 50.1±1.1 | **94.1±0.5** | **64.6±1** | **95.7±0.1** | **74±0.6** | 11.8±0.9 | 37.6±0.3 | **50±1.4** |
| (best layer) | (L9,10,11) | (L5) | (L7) | (L4) | (L7,8,9) | (L9) | (L11) | (L4,7) |
| Fine-Tuned | 50±0.6 | 95.5±0.8 | 91.8±1.2 | 98.7±0.09 | 80.4±0.8 | 12±1.3 | 40.5±0.4 | 91.3±0.5 |

Table 11: Constrained-data task matrices against baselines (%). With training samples limited to 20% of the original quantity, results remain consistent. DeiT-tiny-patch16-224 (n=5, 95% CI). Layers are zero-indexed.

| Method | DTD | EuroSAT | GTSRB | MNIST | RESISC | Stanford Cars | SUN397 | SVHN |
|---|---|---|---|---|---|---|---|---|
| (classes) | (47) | (10) | (43) | (10) | (45) | (196) | (397) | (10) |
| Base w/ FT Classifier | 3.1±0.6 | 14.5±3.9 | 4.2±1.9 | 12.8±3.3 | 2.9±0.3 | 0.6±0.1 | 0.3±0.1 | 11.7±2.5 |
| Task Matrix (best layer) | **59.7±0.5** (L8,9) | **94.6±0.3** (L5) | **63±1.4** (L7) | **95.1±0.4** (L3,4,7) | **75.3±1.2** (L7,8) | **16±1.9** (L9) | **24.7±1.6** (L10) | **48.8±1.2** (L7) |
| Fine-Tuned | 63±1.5 | 98.5±0.2 | 98.6±0.1 | 99.5±0.06 | 90.5±1 | 26.9±5 | 38±1 | 96.1±0.2 |

Table 12: Task matrix performance against vision baselines (%). The classifier head was randomly initialized, frozen while fine-tuning, and used for evaluations. Results remain consistent and approach finetuned levels over all datasets. DeiT-tiny-patch16-224 (n=5, 95% CI). Layers are zero-indexed.

| Method | DTD | EuroSAT | GTSRB | MNIST | RESISC | Stanford Cars | SUN397 | SVHN |
|---|---|---|---|---|---|---|---|---|
| (classes) | (47) | (10) | (43) | (10) | (45) | (196) | (397) | (10) |
| Base w/ FT Classifier | 2.7±0.5 | 12±3 | 4.8±0.8 | 12.8±1.3 | 3.4±0.9 | 0.6±0.09 | 0.3±0.1 | 15±3 |
| Task Matrix (best layer) | **47.2±1.3** (L9,10) | **94±0.1** (L5) | **59.2±1.1** (L7) | **94.9±0.4** (L4) | **72.8±0.3** (L7) | **0.8±0.1** (L3,8,9,11) | **12±1** (L10) | **50.6±1.1** (L7) |
| Fine-Tuned | 30±0.8 | 95.8±0.7 | 93.9±0.9 | 98.8±0.1 | 76.9±1.3 | 0.5±0.1 | 2.6±0.4 | 91.1±0.7 |

Table 13: Task matrix performance against vision baselines (%). The classifier head was randomly initialized, frozen while fine-tuning, and used for evaluations. With training samples limited to 20% of the original dataset, results remain consistent and approach or exceed finetuned levels over all datasets. DeiT-tiny-patch16-224 (n=5, 95% CI). Layers are zero-indexed.

# E DINOv3 ViT-B/16: Single Task Results

We extended the datasets used in order to comprehensively evaluate a novel vision transformer. Specifically, we experiment on INaturalist-Mini 2021 Horn et al. (2021), Cifar10 Krizhevsky & Hinton (2009), Cifar100 Krizhevsky & Hinton (2009), and Food101 Bossard et al. (2014). The task matrix generally exhibits lower results than linear probes. We posit the task matrix's lower performance to DINOv3 ViT-B/16's strong pre-trained backbone, and the lack of middle-layer enrichment in vision models.

| Method | DTD | EuroSAT | GTSRB | MNIST | RESISC | Stanford Cars | SUN397 | SVHN |
|---|---|---|---|---|---|---|---|---|
| (classes) | (47) | (10) | (43) | (10) | (45) | (196) | (397) | (10) |
| Linear Probe | **83.5±0.1** | 97±0.09 | 85.7±0.2 | 98.7±0.03 | **92.7±0.1** | **93.8±0.1** | **77.2±0.09** | **66.9±0.1** |
| Task Matrix (best layer) | 82.9±0.3 (11) | **97.1±0.1** (6,9,11) | **86.3±0.7** (11) | **98.9±0.1** (8) | 91.4±0.1 (11) | 93.7±0.09 (11) | 76.7±0.4 (11) | 63.1±1 (8) |
| Fine-Tuned | 84.5±0.2 | 98.8±0.1 | 98.9±0.1 | 99.5±0.1 | 95.7±0.2 | 94.4±0.09 | 78.1±0.3 | 97±0.1 |

Table 14: Task Matrix against vision baselines (%), DINOv3 ViT-B/16 (n=5, 95% CI). Layers are zero-indexed.

# F Text Dataset descriptions

Emotion (Saravia et al., 2018): A text classification dataset containing English Twitter messages labeled with six basic emotions (anger, fear, joy, love, sadness, and surprise), designed to evaluate models' ability to recognize emotional content in social media text.

| Method | INaturalist-2021 | Cifar10 | Cifar100 | Food101 |
|--------|------------------|---------|----------|---------|
| (classes) | (10000) | (10) | (100) | (101) |
| Linear Probe | **60.9±0.7** | **98±0.01** | **88.5±0.1** | **93.3±0.08** |
| Task Matrix | 59.1±0.6 | 97.7±0.1 | 86.7±0.5 | 92.8±0.2 |
| (best layer) | (11) | (11) | (11) | (11) |
| Fine-Tuned | 68.4±0.9 | 98.9±0.1 | 92±0.5 | 93.4±0.2 |

Table 15: Task Matrix against vision baselines (%), DINOv3 ViT-B/16 (n=5, 95% CI). Layers are zero-indexed.

HANS (McCoy et al., 2019): A diagnostic dataset for natural language inference that systematically tests syntactic heuristics by containing examples where lexical overlap, subsequence, and constituent heuristics fail, revealing models' reliance on spurious statistical patterns rather than genuine linguistic understanding.

BLiMP (Warstadt et al., 2020): The Benchmark of Linguistic Minimal Pairs for English, consisting of 67 sub-datasets with 1,000 minimal pairs each that isolate specific contrasts in syntax, morphology, or semantics, enabling targeted evaluation of models' grammatical knowledge.

TREC-6 (Li & Roth, 2002): A question classification dataset containing 5,500 labeled questions divided into 6 coarse semantic categories (abbreviation, entity, description, human, location, numeric) for open-domain, fact-based question answering systems.

SNLI (Bowman et al., 2015): The Stanford Natural Language Inference corpus containing 570k human-written English sentence pairs manually labeled for entailment, contradiction, and neutral relationships, serving as a foundational benchmark for natural language understanding.

ATIS (Hemphill et al., 1990): The Airline Travel Information Systems dataset consisting of audio recordings and transcripts of humans asking for flight information, containing 17 unique intent categories for evaluating spoken language understanding systems.

Banking-77 (Casanueva et al., 2020): A fine-grained intent detection dataset in the banking domain comprising 13,083 customer service queries labeled with 77 distinct intents, designed to evaluate models' ability to understand specific user intentions in specialized domains.

## G  VISION DATASET HANDLING

For the standard 8 classification datasets used across CLIP ViT B/32 Vision Tower, DeiT-tiny-patch16-224, and DINOv3 ViT-B/16, we utilized the publicly available dataset, "The Eight Image Classification Tasks" (Tangake).

DTD, MNIST, Stanford Cars, and SVHN contain the full number of original dataset images, while SUN397 is the 50-class split for both training and testing partitions. EuroSAT, GTSRB, and RE-SISC45 contain 2,700, 12,569, and 6,300 fewer total images than the full original dataset respectively.

For DINOv3 ViT-B/16, we utilized the full datasets for INaturalist 2021, Cifar10, Cifar100, and Food101.

