# OpenReview forum: "Task Matrices: Linear Maps for Cross-Model Finetuning Transfer"
_ICLR.cc/2026/Conference — Submitted to ICLR 2026_

### Official Review · Reviewer_jcUf · 2025-10-23

**Soundness:** 1
**Presentation:** 2
**Contribution:** 2
**Rating:** 2
**Confidence:** 5

**Summary:**

The work proposes learning a linear map between the representations of the base model and those of its finetunings, and show that this allows obtaining close-to-finetuning performance just by equipping the base with this extra adapter layer. Fitting the map requires task-specific data, and the manuscript show results both using 100% of the task data as well as a fraction (20%), showing a graceful degradation. The proposed approach is compared with linear probes trained with the same data, showing the learned base-to-finetune linear maps to obtain better performance, sometimes getting close to that of the finetuned model. Experiments are performed with RoBERTa and all-MiniLM-L12-v2 for text, and ViT, DeIT and DINOv3 for vision.

**Strengths:**

- Although very simple, the idea of fitting an adapting from the base model to its finetuning makes intuitive sense, and might actually be useful in several practical use cases. Indeed, having a whole finetuning compressed into a single vector is what made task vectors [1] particularly enticing, and in this case the dimension of the task "digest" would be much smaller than that of the whole network.

- The results, if confirmed, seem promising and interesting. To the best of my knowledge, this analysis is also novel. Differences between finetuned and base models have been mostly studied in the weight-space, with this work focusing instead on the representation space.

[1] Ilharco, Gabriel, et al. "Editing models with task arithmetic." The Eleventh International Conference on Learning Representations.

**Weaknesses:**

- The whole “interpretability suggests that vision and language models learn implicit linear encodings”, repeated several times throughout the manuscript as main motivation, is somewhat vague and misleading. What the cited prior work claims is that relation decoding is (sometimes) well approximated by a linear transformation, not that the overall function computed by a transformer block, or a whole transformer, can be approximated thus. In this perspective, approximating a whole finetuning by a linear transformation over some chosen layer seems arbitrary.

- The CLIP experiments employ a trainable classifier instead of doing open vocabulary classification with the frozen text encoder. This is different than what is usually done for open vocabulay classification on multi-modal architectures such as CLIP. This makes it hard to assess how much the finetuned head contributes to the final accuracy.
    - In this matter, I find the results in table 4 and 5 intuitive as the finetuned head expects different statistics than those presented by the activations obtained from the base, but this does not really rule out that the benefit stems from the finetuned head.
    - To actually rule this one out, one would have to use CLIP’s frozen text encoder as in most works. I tried myself implementing the proposed method using CLIP’s text encoder but I obtain results close to random-chance.

- No code is provided. Given the lacking of theoretical justification and the surprising nature of the results, I cannot reliably judge this paper without being able to reproduce its results. To this end, I performed a best-effort implementation of the method with the only difference being the usage of the frozen CLIP text encoder instead of an ad-hoc classifier as done in the paper. I could not reproduce the results; in fact, I could not get anywhere close to the reported scores.

- The merging experiment needs some baselines for comparison. With a quick comparison to recent literature, the method does not seem competitive to merging techniques: the single task matrix in figure 4 obtains ~80% normalized average accuracy, below modern merging techniques which surpass 90% [2, 3].

- Writing and figures seem a bit rushed. There is a dangling acknowledgment section stating “We thank ….”.

Overall, I feel like the idea is interesting and worth exploring in depth, but the uncertainty of the empirical evidence and the overall lack of motivation suggest the paper not to be ready for publication at its current stage.

[2] Gargiulo, Antonio Andrea, et al. "Task singular vectors: Reducing task interference in model merging." *Proceedings of the Computer Vision and Pattern Recognition Conference*. 2025.

[3] Marczak, Daniel, et al. "No Task Left Behind: Isotropic Model Merging with Common and Task-Specific Subspaces." ICML 2025.

**Questions:**

- There is no clear insight on when the method will work and when not. Can you provide experimental evidence or theoretical discussion about when and why to expect it to work in practice? Studying the structure of the learned map may help here. e.g. is it low-rank? is it sparse?

-  Did you try aggregating the linear maps instead of fitting one linear map for all the tasks?

- Why wasn't the CLIP text encoder used? I couldn't get it to work. Are there any reasons why this method should only work with linear classifier heads instead of frozen text encoders in an open vocabulary setting?

---

> ### Author Response · Authors · 2025-11-26
> **Official Response**
>
> Thank you for the in-depth commentary on the soundness of our work. We agree that ‘implicit linear encoding’ is vague and potentially misleading terminology, and take care to qualify the statement where it appears. The cited work identifies a causal relationship for the task of object prediction over relational predicates, carried out primarily by influential middle layers of the object embedding. This work has been generalized for a range of in-context settings [1], motivating the authors to consider relational context as a form of task adaptation. Claiming that the method proposed is approximation by linear transformation from a chosen layer is a simplification– rather, the linearity assumption is for the final finetuned layer, such that mappings optimize for representational alignment with the unembedding.
>
> Seeing the nature of difficulty in producing results for CLIP with the text-encoder model, we ran experiments with fine-tuned models available on HuggingFace for easy reproduction. We provide the following anonymous repository on which frozen-text encoder CLIP experiments can be run: https://anonymous.4open.science/r/ICLR_2026_Submission_13653/.
>
> We report the performance for CLIP with only the vision encoder fine-tuned using models from [2]. These results **demonstrate the task matrix performance is not random, and reaches competitive performance with the proposed settings.**
>
> $$
> \\begin{array}{lcccccccc}
> \\text{Method} &
> \\text{DTD (47)} &
> \\text{EuroSAT (10)} &
> \\text{GTSRB (43)} &
> \\text{MNIST (10)} &
> \\text{RESISC45 (45)} &
> \\text{Cars (196)} &
> \\text{SUN397 (397)} &
> \\text{SVHN (10)} \\\\
> \\hline
> \\text{Base} &
> 40.79 & 42.11 & 27.57 & 30.75 & 51.01 & 57.95 & 60.40 & 14.76 \\\\
> \\text{Task Matrix} &
> 72.48\\,(11) & 96.51\\,(7) & 84.82\\,(11) & 97.34\\,(9) &
> 88.03\\,(11) & 71.50\\,(11) & 69.98\\,(11) & 60.84\\,(8) \\\\
> \\text{Fine-Tuned} &
> 76.38 & 98.66 & 97.69 & 99.32 &
> 94.26 & 76.79 & 71.52 & 95.93 \\\\
> \\end{array}
> $$
>
> CLIP Experiments: Our motivation for employing CLIP with a classifier head was to ensure consistency with text experiments. While we believe the ablations in 5.5 sufficiently demonstrate the Fine-Tuned Classifier head is not contributing to meaningful performance, to fully address the reviewer’s concerns we conducted the experiment above. We conclude that task matrices are causing the increased performance, rather than the finetuned head or frozen encoder.
>
> Implementation Difficulties: See the repository link in the section above.
>
> Merging Experiment Baselines: While existing merge techniques do perform well within the vision setting, **our method is highly competitive with state-of-the-art model merging on text discrimination tasks**. See the general comment.
>
> Writing and Figures: We have removed the “We thank…” portion from the acknowledgements section, and made edits throughout to improve clarity and flow.
>
> Questions:
>
> #1, 2: Task matrices are successful at recovering the majority of performance across all vision and text tasks. in 5.2 ‘Layer by Layer Performance’, we demonstrate that mini-LM and RoBERTA exhibit *qualitatively different behavior* than vision models, and are linearly decodable from non-final representations. We invite the reviewer to revisit sections 5.3 ‘Task Matrices in Data-Scarce Settings’, in which we show the learned mapping is robust to changes in data quantity for both RoBERTa and CLiP, as well as ‘5.4 Task Matrices for Multi-task Classification’. The map learned for 5.4 aggregates all input data from multiple datasets. We did not explore data mixture, which is an interesting direction for further research.
>
> We performed extensive experiments to understand the learned mappings. We found that they are not low-rank, and the rank increases as the layer increases. However, they are sparse, with 2%-69% of the singular values capturing 95% of variance. The number of singular values increases as the number of classes within the dataset increase. Further research could investigate whether these factors are predictive of approximability, or intrinsic properties of learned transformations.
>
> *If the reviewer meant averaging singular task matrices into a single task matrix, we performed 2-task experiments and found results close to near random. There is no reason to expect the learned transformations to be linearly interpolable.
>
> #3: CLIP text Encoder: See the response above. **We show that the method works not only with linear classifier heads, but with frozen text encoders.**
>
> [1] Xia, E., & Kalita, J. (2025). Linear Relational Decoding of Morphology in Language Models. arXiv preprint arXiv:2507.14640. https://arxiv.org/abs/2507.14640
>
> [2] Anke Tang, Li Shen, Yong Luo, Han Hu, Bo Du, and Dacheng Tao. Fusionbench: A comprehensive
> benchmark of deep model fusion, 2024. URL https://arxiv.org/abs/2406.03280.

---

### Official Review · Reviewer_9Wyt · 2025-10-24

**Soundness:** 3
**Presentation:** 3
**Contribution:** 3
**Rating:** 4
**Confidence:** 4

**Summary:**

This paper presents and defines the concept of **Task Matrix**, a linear transformation that maps an intermediate embedding state of a pretrained base model to the corresponding finetuned embedding state. The authors empirically show that such a linear encoding exists between pretrained and finetuned architectures, and that this relationship can be exploited using task matrices as an alternative to linear probing. Experiments on both vision and text tasks demonstrate that a base pretrained model augmented with a task matrix can achieve results that are comparable to, and in several cases surpass, linear probes, sometimes approaching finetuned accuracy. The proposed method is conceptually simple, providing a compact way to transfer finetuned behavior without retraining model parameters.

**Strengths:**

Method simplicity: the proposed task matrix construction method is straightforward and easy to implement. It consists essentially of a least-squares regression that learns a linear map between base and finetuned hidden representations.

Generalization across multiple datasets and models: the approach is shown to generalize well to both vision and text models, and even across multiple datasets simultaneously. The authors demonstrate with extensive experiments that a single task matrix can support multiple classification tasks with only marginal accuracy drops.

Data efficiency and robustness: Task matrices can be trained with very small amounts of data, sometimes below 1% of the training set, while maintaining strong performance with respect to linear probing perfomed on the same amounts of data. This suggests potential advantages in low-data or privacy-restricted scenarios.

**Weaknesses:**

No clear conceptual advantage over existing baselines: while the approach is elegant and achieves competitive performance, its advantages over standard techniques such as linear probing or low-rank adaptation remain unclear in the full-data setting.
The authors describe task matrices as “lightweight” and “low-cost,” but do not provide runtime, FLOPs, or parameter count comparisons against LoRA, adapters, or full finetuning baselines.

Potential equivalence to linear probing under specific conditions: if the optimal base embedding for learning the task matrix W corresponds to the final layer of the base model, the method effectively collapses to standard linear probing, since both rely on the same linear relationship between final-layer embeddings and the output space. Empirically, this appears to be the case for vision tasks, where the best-performing task matrices are consistently derived from the final layer.

Readability and notation issues: although the general concept is clear, the paper suffers from minor inconsistencies in notation and some missing definitions. For example, in Section 3.2 (lines 168–173), the set indices (S) are inconsistently indexed by (n) instead of (k). The overall exposition of the multitask classification setting (Section 5.4) lacks detail on how the base and final layer embeddings are sampled from the joint dataset.

Dependence on fine-tuned model embeddings: the method requires access to the embeddings of the target fine-tuned model in order to construct the task matrix. This dependency substantially limits the practicality and claimed efficiency of the approach, since obtaining such embeddings presupposes the existence of a fine-tuned model. In effect, the method relies on the same model it seeks to approximate or replace, which undermines its utility for scenarios where fine-tuning is computationally infeasible or where fine-tuned weights are unavailable.

Statistical reporting ambiguity: throughout the figures and tables, the notation “n=5, CI=95%” is used. I suppose this is intended to mean that results are averaged over five independent runs (n=5) and reported with a 95% confidence interval, but this is not explicitly stated in the paper.

**Questions:**

1) In multitask training, how are datasets balanced or sampled when constructing a shared task matrix?
2. Can task matrices trained on one task be transferred or composed for related downstream tasks?
3. How does the approach behave when the finetuned model diverges substantially from the base model (e.g., nonlinear adaptation or large domain shift)?

---

> ### Author Response · Authors · 2025-11-26
> **Official Response**
>
> We thank the reviewer for taking the time to review our submission. We are pleased that you find our approach ‘elegant’, and appreciate the recognition of its data efficiency. We address the weaknesses listed below.
>
> No clear conceptual advantage over existing baselines: As stated in our paper, the task matrix is a $d_{\text{embed}}$ x $d_{\text{embed}}$  matrix, and is derived from a least-squares computation mapping $d_\text{embed} \times N$  to $N \times d_\text{embed}$ matrices. Though the parameter count is higher than a linear probe, which is $(\text{num classes}* 768) + \text{num classes}$, the matrix does not go through any epoch of tuning; in fact, the matrix is only constructed once with least squares regression, which only involves SVD for least squares. This means that the technique proposed is guaranteed to be **compute-efficient** compared to existing parameter-efficient methods. On the other hand, the linear layer / classifier must go through multiple epochs of tuning to reach comparable performance.
>
> Our paper focuses on investigating general properties of this transformation, as opposed to explicit runtime comparisons. Please see the general comment for more information.
>
> The equivalence to linear probing under specific conditions is an astute observation. However, mini-LM and RoBERTA results exhibit qualitatively different behavior, and are linearly decodable from non-final representations. RoBERTA results often peak at early layers (1-6) and intermediate layers (11, 16, or 17) (Table 2). Although similar behavior has been observed within in-context learning, we believe the cross-model effect is novel and surprising.
>
> Readability and notation issues. We have made the appropriate corrections.
>
> Dependence on fine-tuned model training: Our method is grounded in a different theoretical approach supplementary to existing parameter-efficient methods such as LoRA and Bitfit, and can be used in conjunction with either method. LoRA requires backpropagation and optimization of adapter matrices, whereas our technique requires only singular-value decomposition of provided samples.
>
> Statistical reporting ambiguity. We have added a note in the results section clarifying this confusion.
>
> Here are answers to the questions:
>
> #1: In Multitask training, the joint dataset is collected over each selected dataset. We did not explore data mixture, which is an interesting direction for further research and could lead to further improvements.
>
> #2: A task matrix is effective for the dataset which is its approximation objective. Task matrices can be composed highly effectively across tasks, as seen in Section 5.4.
>
> #3: The pretrained models exhibit low baseline performance across all tasks: thus, each finetuning dataset involves a large domain shift. However, the linearity hypothesis is predicted on the fine-tuned model having implicit knowledge of domains, accessed through linear transformations of early layer representation. Thus, we view transitioning from pretrained to finetuned models more akin to domain adaptation.

---

> > ### Comment · Reviewer_9Wyt · 2025-11-26
> >
> > I thank the authors for the detailed response and clarifications. While I acknowledge the point that solving for the matrix via SVD is computationally faster than backpropagation, I remain unconvinced about the method's practical utility due to a fundamental circular dependency: constructing the Task Matrix requires target embeddings ($\hat{h}^L$) from a fully fine-tuned model. Thus, the user must incur the substantial cost of fine-tuning *before* employing this method, effectively negating the claimed training advantage over baselines. Furthermore, regarding the lightweight claim, a $d \times d$ Task Matrix (e.g., $\approx 590k$ params for $d=768$) is significantly larger to store and deploy than standard parameter-efficient adapters like LoRA. Consequently, the method offers no storage or inference advantage over baselines while introducing a heavy prerequisite. While the observation regarding intermediate layer linearity is interesting for interpretability, the paper frames this as a novel adaptation technique without establishing a clear practical scenario where it is preferable to existing solutions.

---

> > > ### Author Response · Authors · 2025-12-01
> > > **Official Response**
> > >
> > > We thank the reviewer for their valuable additional feedback. The authors acknowledge that finetuning must occur prior to the construction of task matrices, which indeed introduces a heavy prerequisite. However, they would like to note that the augmentation of existing models with finetuned versions is also a desirable capability, which has been investigated with task arithmetic approaches [1].
> > >
> > > The vision results from 5.4 suggest that task matrices are competitive with such results. We acknowledge that further experiments need to be performed on the text side, and are grateful for the feedback received thus far.
> > >
> > > [1] He, Y., Hu, Y., Lin, Y., Zhang, T., & Zhao, H. (2024). Localize-and-Stitch: Efficient Model Merging via Sparse Task Arithmetic. arXiv preprint arXiv:2408.13656. https://arxiv.org/abs/2408.13656

---

### Official Review · Reviewer_XzQi · 2025-10-30

**Soundness:** 2
**Presentation:** 2
**Contribution:** 2
**Rating:** 2
**Confidence:** 4

**Summary:**

This paper introduces Task Matrices, a simple but intriguing method for transferring the effects of fine-tuning across models via a learned linear transformation between embedding spaces. Specifically, the authors posit that for a base model and its fine-tuned counterpart, there exists a linear mapping $W$ such that $Wx\approx y$, where $x$ and $y$ are hidden representations from the base and fine-tuned models respectively. They show empirically that this assumption holds across multiple architectures (RoBERTa, CLIP ViT, DeiT, DINOv3) and domains (text and vision). The task matrix is learned via regression and can then be used to approximate fine-tuned performance with minimal data and compute cost. Results show that task matrices outperform linear probes and sometimes approach fine-tuned accuracy.

**Strengths:**

**Conceptual novelty:** The central idea -- that fine-tuning can be approximated as a linear transformation in representation space -- is novel and elegant. It reframes fine-tuning as a geometric relation rather than an optimization process, opening potential directions for efficient adaptation and cross-model understanding.

**Empirical coverage:** The authors conduct an extensive empirical study spanning both vision and language domains, multiple model families, and diverse datasets.

**Ablations:** The paper includes ablation studies disentangling the role of the task matrix from classifier heads and fine-tuned weights. These experiments convincingly show that the performance gains come from the learned transformation itself rather than from trivial parameter reuse.

**Weaknesses:**

**Experimental discussion and clarity.** While the empirical coverage is broad, the presentation and discussion of results are weak. Tables are numerous and scattered across the main and supplementary sections, but the commentary does not synthesize the findings into a coherent take-away. For instance, the reader is never clearly told \textit{whether} and \textit{under what conditions} the task matrices work best. A concise summary of results (e.g., performance trends across depth or domains) is missing. Such discussion is critical for a paper that is primarily empirical in nature.

**Practical usefulness:** It remains unclear in what concrete scenarios the method is beneficial. The requirement of having a fine-tuned model to construct the task matrix largely defeats its potential use as a parameter-efficient fine-tuning (PEFT) alternative. It does not reduce the computational cost of fine-tuning, nor does it serve as an effective distillation mechanism — in vision settings, it even increases model size by often appending an additional linear layer. The discussion around multi-task classification could provide a compelling application, but it is presented vaguely and without quantitative results, leaving its value difficult to assess.

**Layer selection ambiguity.** The paper reports results as a function of layer ID, but the criteria for selecting the "best layer" are not properly explained. If the choice is based on validation accuracy, the best-performing layer would trivially tend to be the final one. Moreover, since models of different depths are compared, layer indices are not directly interpretable -- expressing layer position as a fraction of total depth would have been clearer and more comparable.

**Lack of theoretical analysis**: The central linearity assumption, while empirically explored, lacks theoretical grounding. The paper would benefit from a deeper analysis of when and why such linear mappings hold — for example, as a function of layer depth, task similarity, or representation alignment. Without this, the method risks being a descriptive observation rather than a generalizable principle.

**Limited comparisons:** The paper only compares the proposed approach against linear probing, which is a relatively weak baseline, moreover it's not explained why this should be an interesting comparison. To substantiate the claimed advantages, it would be important to include comparisons with stronger and more relevant alternatives — such as modern knowledge distillation or parameter-efficient fine-tuning methods. Without these, it is difficult to assess the practical competitiveness of the proposed method.

**Decontextualized experiments.** Some experimental setups (e.g., data-scarce, multi-task) are introduced without sufficient justification or connection to the paper’s stated goals. After each, the reader is left wondering: "why is this experiment important?" The results are not tied back to an overarching research question or potential application. This gives the impression of a collection of loosely related tests rather than a coherent investigation.

**Writing and presentation.** The exposition requires substantial improvement, especially from Section 5 onward, where the paper becomes quite confusing (for instance, lines 404–408 are unclear). The main claim is buried in the text rather than stated explicitly in the introduction or conclusions. Notation is inconsistent — for example, lines 154–160 introduce symbols in a confusing way, and this notation is never reused later. Equations are unnumbered, and figures and tables are poorly designed (e.g., Figure 2 is almost unreadable in grayscale). Table 4 appears incomplete, as it contains only a single row, and the large number of tables without consistent headers makes it difficult to interpret the results. Overall, the paper would benefit from a clearer structure, stronger motivation, and more precise phrasing.
\end{itemize}

**Minor issues, not influencing the final assessment:**

- Line 40: "offer lightweight and effective approximations", should be "offers" (subject "model states" is singular).
- Line 95–96: "while in text, while in text" repetition.
- Line 122: "have subsequently provided", should be "has subsequently provided" (subject "body" is singular).
- Line 154-155: "be $H^L$ be $H_{ft}$" repetition
- Line 174: Equation brackets misaligned: "V ($\hat{h}^L$))": remove extra parenthesis: "V ($\hat{h}^L$)".
- Line 240-241 "state-of-art" should be "state-of-the-art".
- Line 248 "\textit{eight} diverse NLP benchmarks", only seven text datasets are presented
- Line 260-261 "casual influence" should be "causal influence"

**Questions:**

No questions.

---

> ### Author Response · Authors · 2025-11-26
> **Official Response**
>
> Thank you for taking the time to review our paper and recognizing the novelty of our method. We address the weaknesses below.
>
> Experimental discussion and clarity: We would like to clarify that we do summarize the performance across depths for both text and vision domains in section 5.2, Layer-By-Layer Performance. We clearly state that with text models, performance remains generally stable at intermediate layers, while with vision models, performance is best at the later layers. Moreover, we identify that the greater number of classes there are in vision, the greater the intermediate linearities as evidenced by the sharp increase in accuracy.
>
> Practical usefulness: In Section 1, Introduction, we note that our novel adaptation technique is useful when releasing fine-tuned models is impractical or commercially infeasible. Moreover, our method is a form of self-distillation parameter-efficient fine-tuning (PEFT)--- the fine-tuned and base models are akin to the “teacher” and “student,” respectively. Though Task Matrices do not exactly match the performance of the fine-tuned model, they still demonstrate a generally strong approximation, opening future work in the field to enhance our method. **We have added a multi-task normalized heatmap to the appendix showcasing the performance of 2-task Task Matrices**.
>
> Layer selection ambiguity: We explicitly state the process for finding layers in our tables, being the best layer which translates to best accuracy. **It is not the case that the best-performing layer is the final one**. On the text side, most results show augmented model performance peaking as early as the first layer.
>
> Lack of theoretical analysis: Our paper is grounded in prior literature. Our linearity assumption builds upon established findings that transformers create linear substructures for relational reasoning. (Mikolov et al. 2013) demonstrated linear relationships in word embeddings. (Hernandez et al. 2023) showed that transformer in-context learning brings about linear transformations between related concepts within a transformer’s intermediate layers: We focused on intermediate layers due to the existence of middle-layer MLPs containing enriched representations, and Dai et al. (2022) demonstrated that intermediate representations are more transferable across tasks than final-layer representations. While the issue is subjective, we believe our paper goes beyond descriptive observation to a novel and interesting contribution. Results across vision and text models across fourteen datasets illustrate that task matrices generalize well, are robust to data scarcity, and are grounded in theoretical literature.
>
> Limited Comparisons: See our general response.
>
> Decontextualized experiments: Our evaluation methods follow that of Ilharco et al. (2022), in testing Task Matrices in standard conditions and for multi-task experiments. That being said, we have updated Section 5.3, contextualizing our reason for testing Task Matrices in data-scarce conditions.
>
> Minor Issues: We have made appropriate fixes.

---

> > ### Comment · Reviewer_XzQi · 2025-11-26
> >
> > Thank you to the authors for the thorough response and for the updates made to the manuscript. I appreciate the clarifications provided; several points have indeed been improved, while others still feel only partially addressed.
> >
> > In particular, I believe it is crucial for the paper to be more transparent regarding which layer is selected and which metric is used for this selection. The response mentions “accuracy,” but this choice seems somewhat at odds with the practical motivation emphasized by the authors themselves (namely, enabling a distillation-like process where the trade-off between accuracy and computational cost is more important than accuracy alone). I strongly encourage the authors to rethink and better justify this aspect, possibly considering alternative or complementary criteria that align more closely with the intended real-world utility.
> >
> > More broadly, I suggest carefully integrating the various points raised across the reviews, many of which highlight similar issues regarding clarity, framing, and experimental justification. Addressing them cohesively would significantly strengthen the contribution.
> >
> > That said, I genuinely appreciate the core idea behind Task Matrices; it is interesting and has potential. With further refinement, I believe this work can mature into a valid contribution to the field.

---

### Official Review · Reviewer_5LWE · 2025-10-31

**Soundness:** 2
**Presentation:** 2
**Contribution:** 2
**Rating:** 2
**Confidence:** 4

**Summary:**

This paper introduces the concept of a "task matrix," a linear transformation learned via least-squares regression that maps the intermediate representations of a pretrained (base) model to the final-layer representations of a fine-tuned model. The goal is to create a lightweight and data-efficient method for domain adaptation that avoids the costs of full fine-tuning. The authors demonstrate that for a variety of vision and text datasets, augmenting a base model with a task matrix can outperform linear probing and, in some cases, approach the performance of a fully fine-tuned model. The paper also explores the properties of task matrices, showing they are robust to data scarcity and can be generalized to multitask settings.

**Strengths:**

1. The experiments, conducted across multiple vision (CLIP ViT-B/32) and text (RoBERTa-large) models on ten different datasets, are comprehensive. The results consistently show that the task matrix approach surpasses the performance of linear probing, a standard baseline.
2. A key strength of the proposed method is its performance in data-scarce environments. The experiments show that task matrices maintain a significant performance advantage over linear probes when trained on only 20% of the data.

**Weaknesses:**

1.  **Insufficient Positioning Against Prior Work:** The core idea of using linear maps between embedding spaces has deep roots in prior work on concept learning (e.g., Paccanaro & Hinton, 2001; Mikolov et al., 2013). While the paper applies this to the new context of finetuning transfer, its novelty relative to this established literature is not clearly articulated. Furthermore, the connection to related methods like task arithmetic [Ilharco et al., 2022], which also manipulates model states, is only briefly mentioned and warrants a more thorough discussion to better frame the paper's contribution.

2.  **Mismatch Between Optimization and Inference Objectives:** There is a potential disconnect between the method's training objective and its ultimate goal. The task matrix $W*$ is optimized by minimizing the least-squares error $\|Wh^i - h^f\|^2$, which is the Euclidean distance between the transformed base embedding and the finetuned embedding. However, the inference goal is to match the final classification prediction, i.e., $arg max V(Wh^i) = arg max V(h^f)$. The paper does not provide a theoretical or empirical justification for why minimizing L2 distance in the embedding space is a suitable or optimal proxy for maximizing classification agreement.

3.  **Lack of Guidance on Hyperparameter Selection:** The choice of which intermediate layer from the base model to use is a critical hyperparameter that significantly impacts performance. Figure 2 shows that accuracy can vary dramatically depending on the selected layer (e.g., RoBERTa-SNLI performance fluctuates significantly across layers). The paper offers no principled method, heuristic, or analysis for selecting the optimal layer a priori, which would require an exhaustive and computationally expensive search in practice.

4.  **Limited Comparison to Other PEFT Methods:** The paper's primary baseline is linear probing. While this is a reasonable starting point, the introduction positions the work as an alternative to other prominent parameter-efficient fine-tuning (PEFT) methods like LoRA, BitFit, and Prefix-Tuning. However, no direct comparisons are made. Without this context, it is difficult to assess where task matrices stand in the broader PEFT landscape regarding the trade-offs between adaptation accuracy, memory/storage costs, and computational overhead.

5.  **Unclear and Potentially Incorrect Notation:** Some of the mathematical notation is confusing. For example, the final representation $h^L$ is described as belonging to $H^L \in R^N$, where $N$ is the number of classes. This appears to incorrectly use the number of classes for the hidden dimension, which is defined as $d$ elsewhere. Similarly, the definition of the decoder head as $V ∈ R^{d \times N}$ seems to be transposed from its conventional use in $z = Vh$, where $h$ is $d \times 1$. These inconsistencies make the technical approach harder to follow.

**Questions:**

See weaknesses.

---

> ### Author Response · Authors · 2025-11-26
> **Official Response**
>
> Thank you for the feedback! We are concerned that portions of the review may stem from misunderstandings of our paper, as they make incorrect claims about the scope and substance of the paper. We attempt to clarify the points below, and would appreciate constructive feedback written by the reviewer.
>
> Insufficient positioning against prior work. We mention each of the works listed in the review, and discuss their contributions within Section 1 & 2.1 (Introduction, Related Works).  Our paper demonstrates these linear representations hold between pretrained and finetuned models, within diverse, untested domains for vision and text (including both data-scarce and multitask settings). To the best of our knowledge, the existence of linear encodings across fine-tuning has not been shown before, and is novel and surprising for a general audience.
>
> Mismatch between Optimization and Inference Objectives. As stated in 2.1-2.3 'Related Works' and 3.1 'Linearity Assumption',  **the theoretical case for representational alignment prior to the final classification is well-established**, and has been explored in papers such as [1]. Empirical justification for the specific setup we test can be found in prior work documenting high causality of intermediate layers [2] and enrichment of subject tokens in early MLP layers [3]. Together, these results suggest that models relate early and final internal representations in a linear fashion.
>
> Limited Comparison to Other PEFT Methods. See above.
>
> Unclear and Potentially Incorrect Notation. Thank you for catching this mistake! We have corrected the notation error.
>
> [1] Subramaniam, V., Mayo, D., Conwell, C., Poggio, T. A., Katz, B., Cheung, B., & Barbu, A. (2024). Training the Untrainable: Introducing Inductive Bias via Representational Alignment. CoRR abs/2410.20035.
>
> [2] Hernandez, E., Sharma, A. S., Haklay, T., Meng, K., Wattenberg, M., Andreas, J., Belinkov, Y., & Bau, D. (2024). Linearity of Relation Decoding in Transformer Language Models. In 12th International Conference on Learning Representations (ICLR 2024)
>
> [3] Geva, M., Bastings, J., Filippova, K., & Globerson, A. (2023). Dissecting Recall of Factual Associations in Auto-Regressive Language Models. In Proceedings of the 2023 Conference on Empirical Methods in Natural Language Processing (EMNLP 2023)

---

> > ### Comment · Reviewer_5LWE · 2025-11-26
> >
> > Thank you for the response. Here are my comments to your response:
> >
> > 1. **Positioning against Prior Work:** I acknowledge that some of the works I mentioned were cited; however, I believe the discussion could be more thorough. It appears that you position the paper around the “existence of linear encodings across fine-tuning”. While this is indeed an interesting observation, the paper currently presents it mainly as an empirical finding, without delving into its underlying causes or potential applications. Moreover, given that the proposed method does not perform strongly on certain datasets, framing the contribution solely around this empirical phenomenon feels insufficient.
> >
> > 2. **Mismatch between Optimization and Inference Objectives:** If you consider the theoretical case for representational alignment prior to the final classification to be well-established, it would be best to provide a theorem in the paper showing that these two optimization objectives are equivalent. I am familiar with the cited work, *Linearity of Relation Decoding in Transformer Language Models*. To my knowledge, that work applies to simple relational datasets (e.g., data in the form of “The capital of France is [Paris]”) and does not necessarily hold for general tasks. The gap between relational tasks and the datasets used in your work needs to be considered.
> >
> > 3. **Others:** Some of your comments are missing.
> >
> > Finally, thank you for your response, and I am open to any further discussions.

---

> > > ### Author Response · Authors · 2025-12-01
> > > **Official Response**
> > >
> > > The theoretical justification for our work is as follows. 'Linearity of Relation Decoding' identifies causal relationships for the task of object prediction over relational predicates. This notion of approximation has been shown to generalize to a range of in-context settings (including morphology and lexical semantics) [1]. Because in-context learning is one simple task-adaptation method, it motivates studying fine-tuning from the same perspective, in which task adaptation is achieved by explicitly modifying the model’s internal parameters.
> > >
> > > Mismatch between Optimization and Inference Objectives: The proposed method is not a direct linear transformation from a chosen layer -- the linearity assumption is to the final finetuned layer, such that mappings optimize for representational alignment with the unembedding. Consequently, as motivated by Theorem 2.2 ‘Measurement Representation’ in [2], representation in the unembedding is a natural choice for linear representation of concepts.
> > >
> > > Limited Comparison to Other PEFT Methods:
> > > We first consider the compute costs of the approach. The task matrix is a single $d_{\text{embed}} \times d_{\text{embed}}$ linear operator. Given embeddings $X \in \mathbb{R}^{N \times d_{\text{embed}}}$ and targets $Y \in \mathbb{R}^{N \times d_{\text{embed}}}$, we compute $W \in \mathbb{R}^{d_{\text{embed}} \times d_{\text{embed}}}$ by solving the least-squares problem $XW \approx Y$ via SVD. This requires no gradient updates, no epochs, and no forward–backward passes through the model. LoRA reduces the number of tuned parameters but still relies on backpropagation and computation of activations across the model, so its cost scales with optimization steps and model size. In contrast, our method is a single closed-form solution, making it more computationally efficient than PEFT approaches (including LoRA) and gradient-trained linear heads.
> > >
> > > Performance-wise, our approach is competitive with leading model-merging techniques on text tasks. Dataless Localize-and-Stitch reports 78% recovery on SNLI with RoBERTa-base, whereas our task matrix obtains 86% with RoBERTa-Large. On TREC, Localize-and-Stitch achieves 49% recovery, while our method reaches 89% on the same task. These results suggest that task matrices, which employ linear decoding to the unembedding, are a viable alternative to weight-based model merging [3]. Contrary to 9Wyt’s note on practical limitations due to the necessity of obtaining a finetuned model, augmenting models via merging with finetuned versions is clearly desirable across a range of applications. As seen with the vision multi-task performance, the technique retains performance over multiple datasets. We acknowledge that further experiments need to be performed on the text side and are grateful for the feedback received thus far.
> > >
> > > [1] Xia, E., & Kalita, J. (2025). Linear Relational Decoding of Morphology in Language Models. arXiv preprint arXiv:2507.14640. https://arxiv.org/abs/2507.14640
> > >
> > > [2] Park, K., Choe, Y. J., & Veitch, V. (2023). The Linear Representation Hypothesis and the Geometry of Large Language Models. arXiv preprint arXiv:2311.03658. https://arxiv.org/abs/2311.03658
> > >
> > > [3] He, Y., Hu, Y., Lin, Y., Zhang, T., & Zhao, H. (2024). Localize-and-Stitch: Efficient Model Merging via Sparse Task Arithmetic. arXiv preprint arXiv:2408.13656. https://arxiv.org/abs/2408.13656

---

### Author Response · Authors · 2025-11-26
**General Response: Addressing PEFT & Merge Baselines**

We thank the reviewers for all their feedback! In addition to the general comment, we have released detailed individual responses.

Several reviewers mentioned that there was limited comparison to PEFT techniques. Our primary contribution is not to propose an advance in parameter-efficient fine-tuning, but to study a novel application of a linear transformation within pre-trained architectures. However, we also believe our approach to be more efficient than LoRA and other parameter-efficient training methods.

As stated in the paper, our task matrix is a single $d_{\text{embed}} \times d_{\text{embed}}$ linear operator. Concretely, given an embedding matrix $X \in \mathbb{R}^{N \times d_{\text{embed}}}$ and targets $Y \in \mathbb{R}^{N \times d_{\text{embed}}}$, we compute a matrix $W \in \mathbb{R}^{d_{\text{embed}} \times d_{\text{embed}}}$ that solves a least-squares problem $XW \approx Y$. This is done once, via SVD, and requires no gradient-based optimization and no training epochs.

LoRA reduces the number of trainable parameters by injecting low-rank matrices into each layer, but it is still trained with backpropagation over multiple steps/epochs and requires full forward and activation-gradient computation through the base model. LoRA is more efficient than full fine-tuning, but it is still an iterative training procedure whose cost scales with the number of optimization steps and the full model size. This makes our approach more compute-efficient than PEFT methods including LoRA, and gradient-trained linear heads, which must be iteratively tuned to reach comparable performance.

Because our intervention is a full $d_{\text{embed}} \times d_{\text{embed}}$ matrix mapping from pretrained to fine-tuned representations, the natural comparison is a linear (affine) probe from a single naïve layer representation. Sections 5.2–5.4 are explicitly focused on stress-testing the generalization of these approximations, following conventions established by weight arithmetic papers [1]. While not the focus of the paper, we believe our results are competitive with state-of-the-art techniques from model merging on text discrimination tasks. The sparse arithmetic approach Dataless Localize-and-Stitch [2] achieves 78% recovery on SNLI with RoBERTa-base (Appendix A), for which the task matrix achieves 86% recovery with RoBERTa-Large. Similarly, on TREC, Dataless Localize-and-Stitch achieves 49% recovery with RoBERTa-base, whereas the task matrix achieves 89% recovery. This positions linear layer decoding as a viable alternative model merging technique.

As with the line of work in [1], our method provides an efficient and effective way to transfer capabilities across models. We invite reviewers to revisit the results in the latter half of the results section, in particular §5.3 (“Data-scarce settings”), where the strong baseline of linear probing is substantially outperformed by task matrices.

[1]   Ilharco, G., Ribeiro, M. T., Wortsman, M., Gururangan, S., Schmidt, L., Hajishirzi, H., & Farhadi, A. (2023). Editing Models with Task Arithmetic. CoRR abs/2212.04089 .

[2] He, Y., Hu, Y., Lin, Y., Zhang, T., & Zhao, H. (2025). Localize-and-Stitch: Efficient Model Merging via Sparse Task Arithmetic. arXiv preprint arXiv:2408.13656.

---

### Meta-Review · Area_Chair_Wku1 · 2026-01-07

**Summary:**

This paper proposes Task Matrices, a simple but intriguing method for transferring the effects of fine-tuning across models via a learned linear transformation between embedding spaces. The four reviewers are all negative about this paper. They pointed out multiple critical concerns regarding different aspects of this paper, including but not limited to:
1. The novelty is limited. This paper applies the idea of using linear maps between embedding spaces to the new context of finetuning transfer, but the novelty relative to established literature is unclear. The comparison with related methods remains simple and needs a thorough discussion.
2. There is a potential disconnect between the method's training objective and its ultimate goal. The paper does not provide a theoretical or empirical justification for why minimizing L2 distance in the embedding space is a suitable or optimal proxy for maximizing classification agreement.
3. How the hyperparameters (e.g., the choice of which intermediate layer from the base model to use) are selected lacks guidance. There is no principled method, heuristic, or analysis for selecting the optimal layer a priori, which would require an exhaustive and computationally expensive search in practice.
4. The compared baselines are out of date. Other prominent parameter-efficient fine-tuning methods like LoRA, BitFit, and Prefix-Tuning, should be compared. In the current form, it is difficult to assess where task matrices stand in the broader PEFT landscape.
5. The notations are unclear and probably incorrect. Some of the mathematical notation is confusing.
6. The presentation and discussion of results are weak. Tables are numerous and scattered across the main and supplementary sections, but the commentary does not synthesize the findings into a coherent take-away.
7. It remains unclear in what concrete scenarios the method is beneficial. It does not reduce the computational cost of fine-tuning, nor does it serve as an effective distillation mechanism.
8. The criteria for selecting the "best layer" are not properly explained. Expressing layer position as a fraction of total depth would have been clearer and more comparable.
9. Some experimental setups (e.g., data-scarce, multi-task) are introduced without sufficient justification or connection to the paper’s stated goals.
10. The presentation requires substantial improvement.

The current version needs a thorough revision and cannot be accepted.

**Reviewer Concerns:**

The 3rd, 5th, 7th, 8th, 9th concerns are partially addressed, while the others are still outstanding.

**Reviewer Scores:**

None.

---

### Decision · Program_Chairs · 2026-01-26

Reject